# Candidemia among Hospitalized Pediatric Patients Caused by Several Clonal Lineages of *Candida parapsilosis*

**DOI:** 10.3390/jof8020183

**Published:** 2022-02-12

**Authors:** Rasmus Krøger Hare, Amir Arastehfar, Søren Rosendahl, Arezoo Charsizadeh, Farnaz Daneshnia, Hamid Eshaghi, Hossein Mirhendi, Teun Boekhout, Ferry Hagen, Maiken Cavling Arendrup

**Affiliations:** 1Unit of Mycology, Statens Serum Institut, DK-2300 Copenhagen, Denmark; maca@ssi.dk; 2The Institute for Cancer and Infectious Diseases, Center for Discovery and Innovation, Hackensack Meridian Health, Nutley, NJ 07110, USA; amir.arastehfar@hmh-cdi.org (A.A.); farnaz.daneshnia@gmail.com (F.D.); 3Department of Biology, University of Copenhagen, DK-2200 Copenhagen, Denmark; soerenr@bio.ku.dk; 4Immunology, Asthma and Allergy Research Institute, Tehran University of Medical Sciences, Tehran 14496-14535, Iran; charsizadeh53@gmail.com; 5Division of Infectious Diseases, Department of Pediatrics, Tehran University of Medical Sciences, Tehran 14496-14535, Iran; hamideshaghi@yahoo.com; 6Department of Medical Parasitology and Mycology, School of Medicine, Isfahan University of Medical Sciences, Isfahan 81746-73461, Iran; s.h.mirhendi@gmail.com; 7Reference Mycology Laboratory, Isfahan University of Medical Sciences, Isfahan 81746-73461, Iran; 8Westerdijk Fungal Biodiversity Institute, 3584 CT Utrecht, The Netherlands; t.boekhout@wi.knaw.nl (T.B.); f.hagen@wi.knaw.nl (F.H.); 9Institute of Biodiversity and Ecosystem Dynamics, University of Amsterdam, 1012 WX Amsterdam, The Netherlands; 10Department of Medical Microbiology, University Medical Center Utrecht, 3584 CX Utrecht, The Netherlands; 11Department of Clinical Microbiology, University Hospital Rigshospitalet, DK-2100 Copenhagen, Denmark; 12Department of Clinical Medicine, University of Copenhagen, DK-2200 Copenhagen, Denmark

**Keywords:** candidemia, *Candida parapsilosis* species complex, genotyping, AFLP, microsatellites

## Abstract

*Candida parapsilosis* is the second most common cause of candidemia in some geographical areas and in children in particular. Yet, the proportion among children varies, for example, from 10.4% in Denmark to 24.7% in Tehran, Iran. As this species is also known to cause hospital outbreaks, we explored if the relatively high number of *C. parapsilosis* pediatric cases in Tehran could in part be explained by undiscovered clonal outbreaks. Among 56 *C. parapsilosis* complex isolates, 50 *C. parapsilosis* were genotyped by Amplified Fragment Length Polymorphism (AFLP) fingerprinting and microsatellite typing and analyzed for nucleotide polymorphisms by *FKS1* and *ERG11* sequencing. AFLP fingerprinting grouped Iranian isolates in two main clusters. Microsatellite typing separated the isolates into five clonal lineages, of which four were shared with Danish isolates, and with no correlation to the AFLP patterns. *ERG11* and *FKS1* sequencing revealed few polymorphisms in *ERG11* leading to amino-acid substitutions (D133Y, Q250K, I302T, and R398I), with no influence on azole-susceptibilities. Collectively, this study demonstrated that there were no clonal outbreaks at the Iranian pediatric ward. Although possible transmission of a diverse *C. parapsilosis* community within the hospital cannot be ruled out, the study also emphasizes the necessity of applying appropriately discriminatory methods for outbreak investigation.

## 1. Introduction

In a recent study of the epidemiology of pediatric candidemia in Tehran, Iran, *Candida parapsilosis* complex species isolates accounted for 24.7% of the cases [1]. *C. parapsilosis* is the second most common species in children although with some geographical variation [2,3]. However, *C. parapsilosis* is also the dominating *Candida* species on skin and therefore associated with outbreaks [4,5,6]. Fluconazole-resistant *C. parapsilosis* isolates have been increasingly reported in several countries, including but not limited to South Africa [7], India [8], South Korea [9], Kuwait [10], Mexico [11], Italy [12], and Finland [13], and has been associated with poor outcome and clinical failure [6,14,15]. Outbreaks involving fluconazole-resistant *C. parapsilosis* include azole-naïve patients and limit the use of fluconazole as the first-line agent in developing countries [3,15,16]. Outbreaks may be due to either a common reservoir or horizontal transmission of the pathogen. A recent study from Brazil found isogenic fluconazole-resistant *C. parapsilosis* isolates from the hands of healthcare workers, inanimate surfaces, and patient bloodstream infections [15].

Typing of isolates from patients and hospital environmental screening is essential in outbreak investigations. Amplified Fragment Length Polymorphism (AFLP) fingerprinting and microsatellite typing (Short Tandem Repeat, STRs) have been used as the most common typing tools. A recent study suggested that the latter genotyping method provided a better resolution compared to AFLP fingerprinting [17]. Nonetheless, further assessment is warranted to substantiate this finding. In addition, sequencing of drug target genes (primarily *ERG11*), have previously contributed to delineate clonal outbreaks [6,11,18]. We thus sequenced drug target genes (*FKS1* and *ERG11*) and performed genotyping of the Iranian isolates using AFLP fingerprinting and STR typing.

In comparisons with other genetic markers, AFLP have led to inconsistent results and lack of reproducibility [19]. As AFLP uses dominant markers, the information from heterozygosity is lost, and in studies of outbreaks using AFLP markers, the grouping is often based on “unweighted pair group method with arithmetic mean” (UPGMA) as the clustering algorithm, which is insufficient for analyses of population structure. The codominant STR markers allow the use of *F*-statistics and similar approaches to reveal population structure. Codominant markers can also be analyzed with powerful Bayesian clustering methods, such as STRUCTURE [20], but they often rely on assumptions, such as Hardy–Weinberg equilibrium. These assumptions are clearly violated with organisms that have clonal reproduction. *C. parapsilosis* has been considered an asexual organism [21]. Although evidence has been presented that it may also recombine sexually or parasexually [22], it remains uncertain to what extent mating and recombination occur. Population structure of potential clonal or partly clonal populations can be evaluated by discriminant analysis of principal components (DAPC). The advantage of this approach is that it is based on a prior principal component analysis (PCA) that transforms allele information to uncorrelated variables. This allows analyses of populations in linkage disequilibrium where the markers potentially could be correlated [23].

The purpose of this study was to determine if the high percentage of *C. parapsilosis* belong to a single clonal lineage and are therefore likely part of a hospital-associated outbreak. This was achieved by applying DACP to SSR data. Furthermore, the study compared genotyping by STR and AFLP markers to clarify which of the two typing methods was more appropriate [14].

## 2. Materials and Methods

Isolates. A set of 50 *C. parapsilosis* and six *C. orthopsilosis* isolates from 42 and five candidemic pediatric patients, respectively, hospitalized in Tehran during July 2014–December 2017 were included. The clinical data (Appendix A) and antifungal susceptibility data for these isolates have been described previously [1,24]. Six isolates from the previous study [1] were not stored and omitted for this study. Consecutive isolates from the same species and patient recovered more than 30 days apart were considered as separate episodes occurring in that patient. In addition, 34 isolates from 33 Danish (DK) patients were included as comparators in the microsatellite typing.

Antifungal susceptibility testing was previously performed using the EUCAST E.Def 7.3 method as described in Mirhendi et al. [1].

AFLP fingerprint analysis. The genotypic diversity of isolates was investigated by AFLP as previously described [25]. AFLP data were analyzed by BioNumerics software v7.6 (Applied Math, Sint Martems-Latum, Belgium). The reference and type strains of *C. parapsilosis* (CBS 604, CBS 1818, CBS 1954, CBS 2195, and CBS 2917), *C. metapsilosis* (CBS 2315, CBS 2916, and CBS 10907), and *C. orthopsilosis* (CBS 10906) were included.

Microsatellite typing. Six short tandem repeat (STR) markers were used for microsatellite typing of *C. parapsilosis* sensu stricto as previously described [26]. Two multiplex PCR reactions were run to amplify the trinucleotide (3A, 3B, and 3C) and hexanucleotide markers (6A, 6B, and 6C), respectively. Forward amplification primers were 5′ labelled with FAM, HEX, and TAMRA as described [26], and all primers were acquired from TAG Copenhagen A/S, Copenhagen, Denmark. Following modifications were applied. PCR was run in 25-µL reaction volumes with 2 µL genomic DNA, 0.4-µM primer concentrations, and 1 × Extract-N-Amp™ PCR ReadyMix (from Sigma (now Merck), Søborg, Denmark). PCR amplification was run with 35 cycles and T_M_ at 54 °C. Amplicons were diluted 35× with distilled H_2_O and 1 µL diluted amplicons mixed with 12.2 µL distilled H_2_O and 0.8 µL GeneScan™ 500 ROX™ size standard (Thermo Fischer Scientific, Roskilde, Denmark). DNA fragments were denatured at 95 °C for 1 min, followed by rapid cooling on ice before injection in the genetic analyzer ABI3500xL Genetic Analyzer (Thermo Fischer Scientific (Applied Biosystems), Nærum, Denmark). For all six markers, at least three samples with different sizes (except 6B, which only covered two fragment sizes) were amplified with unlabeled forward primers and subjected to Sanger sequencing (Macrogen, Amsterdam, Holland) in order to correlate fragment size to repeat numbers. 

*ERG11* and *FKS1* sequencing of the *C. parapsilosis* complex isolates were done as previously described [17].

Statistical analysis. The STR dataset was analyzed using the package poppr v. 2.9.3 [27] in R v. 4.0.5 (R Core Team, Vienna, Austria). A genotype accumulation curve was plotted to ensure that the number of loci captured the variation present in the two populations. The mlg.filter function was activated to determine the true number of multi-locus genotypes (MLGs) by using a genetic distance threshold determined using the cutoff predictor tool, which finds a gap in the distance distribution. The analyses carried out in poppr included a summary of diversity measures, Bruvo’s genetic distance [28], and minimum spanning tree based on the distance matric.

The index of association (*I_A_*) was calculated in poppr to test for linkage disequilibrium based on a clone-corrected dataset. The modification of *I_A_* that removes the bias of sample size $\bar{r}$*_d_* [29] was also calculated.

*F*-statistics were calculated using the GenAlEx. Overall *F_ST_, F_IS_*, and *Ht* (gene diversity) were calculated as well as population differentiation *R_ST_*, which is based on a stepwise mutation model [30]. The significance of the *R_ST_* structure between populations was tested using 1000 permutations.

DAPC was run using the adegenet package 2.1.15 (updated 2020) [23]) in R. DAPC analyses were only conducted with de novo grouping, as a priori groupings based on insignificant *R_ST_* values between the two geographical samplings made little sense. Instead, the find.clusters() function was used to determine the number of groups (*K*) de novo, with optimal *K* selected as that with the lowest BIC value. The optimal number of PCs to use in the DAPC was determined using the optim.a.score() and xvalDapc() commands and 1000 replicates. Root mean squared error by number of PC of PCA s (RMSE) was chosen as the most important criterion for selecting the number of PC’s.

Ethical considerations. This study was reviewed by ethical committee members of Tehran University of Medical Sciences and granted with the approval ethical code (IR NIMAD REC 1396 245). Written consent forms were obtained from patients involved in the candidemia project, and patients were assigned with numerical codes for anonymity purposes.

## 3. Results

A total of 50 *C. parapsilosis* and 6 *C. orthopsilosis* isolates from 47 candidemic pediatric patients were included. The median age was 10.5 months (range 2 days to 10 years), and predisposing factors included exposure to vancomycin and third-generation cephalosporins, *N* = 47 (100%); admission at intensive care unit (ICU), *N* = 34 (74%); central venous catheter (CVC), *N* = 41 (89%); total parental nutrition (TPN), *N* = 26 (57%); and underlying conditions, such as prematurity, *N* = 12 (26%); cancer, *N* = 8 (17%); and metabolic disease, *N* = 5 (11%) (Appendix A). All five patients with *C. orthopsilosis* infections survived, while 46% (19/41) of *C. parapsilosis* infected patients succumbed (*p =* 0.067).

AFLP fingerprint was first performed and provided two clusters of *C. parapsilosis* (Table 1 and Appendix A): G1-P covered 44 isolates from 36 patients, and G2-P covered 6 isolates from five patients. Likewise, two clusters were defined for *C. orthopsilosis*: G1-Orth and G2-Orth covering three isolates each from three and two patients, respectively. Four patients (IR-Pt-2, IR-Pt-29, IR-Pt-32, and IR-Pt-39) had two or more isolates that belonged to the same AFLP groups (G1-P, G2-P, G1-P, and G1-Orth, respectively). Eight reference strains were included in the AFLP fingerprint testing, including seven *C. parapsilosis* and one *C. orthopsilosis*. Of these, three *C. parapsilosis* (CBS 1818, CBS 1954, and CBS 604) belonged to G1-P, four *C. parapsilosis* (CBS 2197, CBS 2915, CBS 2194, and CBS 2196) belonged to G2-P, and finally, one *C. orthopsilosis* (CBS 10906) differed from the included Iranian *C. orthopsilosis* isolates.

Microsatellite typing revealed 29 unique genotypes (MLGs) among the Danish isolates of *C. parapsilosis* and 41 (MLGs) among the Iranian isolates (Figure 1 and Appendix A). A genotype accumulation (saturation) curve of MLGs showed sufficient STR markers (six loci) had been included (Appendix A). The calculated *D* value according to Simpson’s index of diversity was 0.992. No apparent correlation was observed between AFLP fingerprinting and microsatellite typing.

Ten isolates were in three clusters each, consisting of 2–5 repeat isolates from the same patient (Appendix A). Twelve isolates were in six clusters (IR-C1 to IR-C6) (16.2%), each consisting of two isolates from different patients. Thus, 12 out of 43 isolates from different patients were in clusters (27.9%). Of note, one Iranian cluster involved patients hospitalized in the same ward and during the same time period. In comparison, the 34 Danish *C. parapsilosis* isolates represented 29 unique genotypes, of which three were clusters: one with two isolates from the same patient and two (6.9%) consisting of three isolates from different patients. Neither the number of clusters among unique genotypes nor the number of isolates (from different patients) in clusters differed between Danish and Iranian isolates (*p =* 0.45 and *p* = 0.42, respectively).

Analyses of STR data revealed no population differentiation between the Danish and the Iranian isolates (*R_ST_* = 0.011, *p* = 0.14), and analysis of molecular variance (AMOVA) showed 99% variation within populations and only 1% among populations (Table 2). The lack of differentiation was also illustrated in the minimum spanning network (MSN) based on Bruvo’s distance, where no structure that could separate the two samplings (DK and Iran) was seen (Figure 1). The network showed no reticulations indicating lack of recombination. This was also revealed by index of association IA and the unbiased index of association *$\bar{r}$_d_*, which gave values significantly different from what is expected of a freely recombining population (*p* < 0.001) (Appendix A).

The Danish and the Iranian isolates had similar genetic diversities. Bruvo’s distance was 0.3898 for the Danish population and 0.3706 for the Iranian. Both values differed significantly from randomized data, again supporting that clonal reproduction is dominating.

The find.clusters in adegenets DAPC revealed K = 5 without a priori groupings, and eight principle components (PCs) were retained for analysis based on the lowest root mean squared error (RMSE). The scatter plot assigned all individuals to the five clusters with high certainty (Figure 2). Most isolates were found in two clusters: cluster 2 and cluster 5, which contained both Danish and Iranian isolates. One cluster (cluster 3) was only found in Iran (Figure 3).

All isolates were susceptible to azoles, echinocandins, and amphotericin B and were *FKS1* (hot-spot) wild-type (Appendix A). Thirty-eight isolates were *ERG11* wild-type, and twelve isolates (24%) had one or two homozygous mutations leading to the following amino acid changes: D133Y, Q250K, I302T, and R398I. All six *C. orthopsilosis* isolates were azole and echinocandin susceptible and displayed wild-type *ERG11* and *FKS1* (hot-spots) sequences (Appendix A).

## 4. Discussion

Previous studies have reported a relatively high incidence of *C. parapsilosis* infections in the children’s ICU at Teheran University of Medical Sciences [1,24]. Since this opportunistic pathogen is known to cause hospital-associated outbreaks [31], this study applied genotyping to determine the extent of clonality among the isolates and help to clarify the possible occurrence of outbreaks in an Iranian pediatric setting. Indeed, AFLP fingerprinting initially suggested two potential clonal clusters, namely G1-P and G2-P, which could have prompted presumptive conclusions possibly due to poor infection control and a critical nosocomial clonal spread.

The results of the microsatellite analyses showed that the Iranian outbreak, which was considered as two clones when assayed with AFLP markers, consisted of five clusters. This can be explained by the low resolution of the current format of the AFLP that introduce null-alleles resulting in binary data. Moreover, the mutation model behind AFLP markers is based on frequency of mutations in restriction sites, whereas microsatellites have a much higher mutation rate, and a stepwise mutation model that allows inclusion of repeat length. This conclusion was supported by a previous study [14], which found an insufficient resolution by AFLP and suggested that STR should be the preferred method for genetic analysis to uncover outbreaks. Nevertheless, it remains to be understood if inclusion of additional primers targeting more restriction recognition sites and/or use an electrophoretic system capable of providing higher-resolution gel pictures would improve the performance of AFLP.

Four out of five clusters were shared by Denmark and Iran. This is in agreement with other studies showing no geographic differentiation of *C. parapsilosis* between France and Uruguay [32] and illustrates the dispersal potential of the fungus. One cluster was only found in Iran, indicating that new clusters may still occur locally.

The clonal nature of the pathogen is obvious from the analyses of *I_A_* and *$\bar{r}$_d_*. Though the DACP analysis cannot distinguish clonal population structures from recombining, the minor overlap between clusters and little uncertainty in cluster assignment indicate clonality. The clonal diploid reproduction of the fungus will tend to increase the genetic differentiation among populations compared to the parent population [21,33], as known from parasite populations. This does not mean that the fungus is not able to undergo recombination. The five clusters may well represent the outcome of genomic recombination, which have also been shown by genomic studies where the genomes of *C*. *parapsilosis* strains showed signs of recombination events [22].

We previously documented that an amino acid substitution in the HS1 of Fks1 was found in an echinocandin-susceptible *C. glabrata* isolate [34] and later identified fluconazole-susceptible *C.*
*parapsilosis* isolate harboring Y132F in Erg11 (unpublished data). Therefore, despite the lack of fluconazole and echinocandin resistance in the current study, sequencing of *ERG11* and *FKS1* was included in order to further delineate molecular correlation between the isolates and to map the genes in susceptible strains. *FKS1* profiles were highly conserved, while *ERG11* showed common variants in more than one-sixth of the isolates, which did not influence azole susceptibilities [1]. Furthermore, these variations (instead of a single variant) did not support the hypothesis of a clonal outbreak as opposed to the previously described Brazilian outbreak with a fluconazole-resistant Erg11 Y132F mutant strain [31]. Microsatellite typing was introduced, and in contrast to the AFLP data, the results showed a high genotypic diversity among Iranian isolates. More clusters were found in the Iranian population compared to the Danish, but this difference did not reach statistical significance. Thus, no clear evidence for a clonal outbreak was found as cause of the high *C. parapsilosis* proportion in the Iranian study population compared to pediatric candidemia in Denmark. Different levels of (hand) hygiene could be significant for the incidence of *C. parapsilosis* infections, and nosocomial infections remain a threat but not necessarily with the same strain. Additional sampling of other patients, hospital workers, and units combined with genotyping of *C. parapsilosis* positive samples could have aided a more detailed evaluation of potential transmission routes. Still, the source of candidemia is primarily regarded as a patient’s own microbiota underlining the key role of hygiene to avoid infection. Moreover, geographical differences in the *Candida* species distribution of the normal colonizing mycobiota are poorly investigated. Therefore, it remains unknown whether the difference in the *C. parapsilosis* proportion in Iranian and Danish pediatric candidemia is due to differences in colonization of healthy people or due to differences in infection control practices.

This study is associated with limitations. As the study was based on a single hospital, the data may not be fully representative of *C. parapsilosis* blood stream infections in the Iranian pediatric population. Moreover, the number of isolates in both groups was limited. The Danish isolates were from the entire country and thus included several hospitals and patients without any likely risk of transmission. Nevertheless, we found that two major clusters were shared between Denmark and Iran. It is likely that if whole genome sequence-based analysis had been adopted, some of these apparent clusters might consist of unique isolates although this would not have altered the main conclusions of this study.

In conclusion, our study rejected the hypothesis of an outbreak at the Teheran pediatric ICU department caused by two clones and underlined that AFLP fingerprinting alone can potentially lead to inaccurate conclusions of clonal outbreaks and should always be supported by other more discriminatory methods.

## Figures and Tables

**Figure 1 jof-08-00183-f001:**
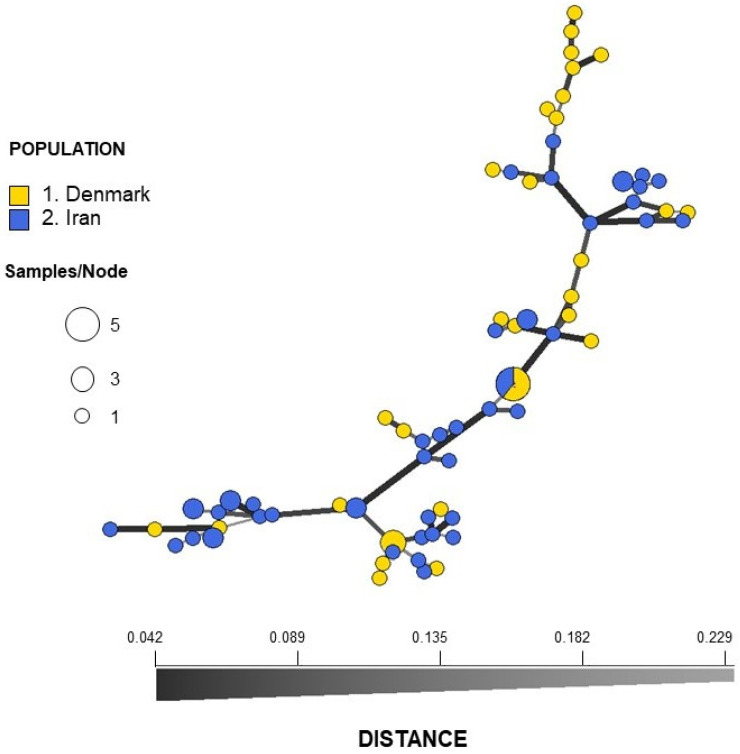
Minimum spanning network (MSNs) inferred using STRs from Danish (blue) and Iranian (yellow) isolates of *Candida parapsilosis*. The network is based on Bruvo’s distance matrix. Each node represents a multi-locus genotype (MLG), with variable size depending on the number of individuals within that MLG. The distance between the nodes represents the genetic distance between MLGs.

**Figure 2 jof-08-00183-f002:**
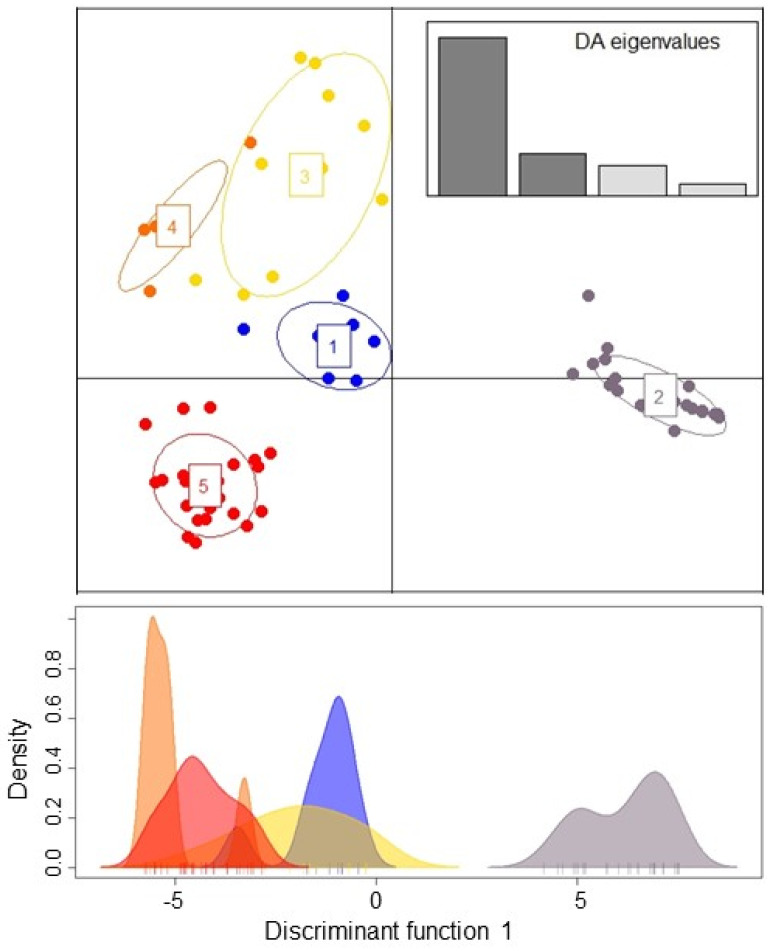
DAPC of Danish and Iranian individuals of *Candida parapsilosis* based on STR data. Scatter plot represents the distribution of individuals (dots) among the five clusters (top). A single discriminant function plotted along the *x*-axis (bottom).

**Figure 3 jof-08-00183-f003:**
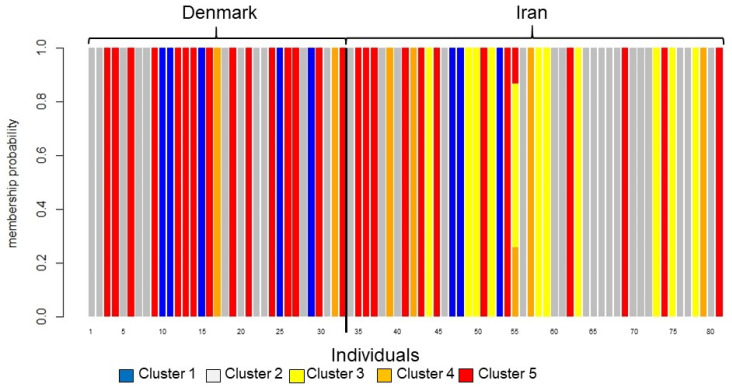
Graphical translation of membership probabilities to the five clusters. The colors of the bars correspond to the clusters in Figure 1. The vertical axis gives membership probability from 0 to 1. All isolates except isolate 55 are assigned with 100% certainty. The black vertical line separates Danish and Iranian isolates. The clusters are almost evenly distributed between Denmark and Iran except cluster 3 (yellow), which is only found in Iran.

**Table 1 jof-08-00183-t001:** Overview of the number and origin of *C. parapsilosis* complex isolates in each AFLP fingerprint cluster (further details are provided in Appendix A).

Cluster	No. Isolates	No. Patients	Wards Involved	Time Span of Obtained Isolates	Reference Strains in Cluster, Origin
G1-P	44	36	12 (PICU, NICU, NICU-OH, CICU, EICU, Cardiac, Neurology, Surgery, Immunology, Infectious, GI, BMT)	2 years and 9 months	CBS 1818, CL CBS 1954, IT CBS 604, PR
G2-P	6	5	4 (NICU, EICU, PICU, Surgery)	2 years and 3 months	CBS 2197, DK CBS 2915, NO CBS 2194, AT CBS 2196, DOM
G1-Orth	3	3	3 (Immunology, PICU, NICU)	1 month	CBS 10906
G2-Orth	3	2	1 (PICU)	10 months	No

G1-P and G2-P for *C. parapsilosis* and G1-Orth and G2-Orth for *C. orthopsilosis,* respectively. Reference strains (CBS) were included and origins provided as land codes. Wards covered: ICU, intensive care unit; PICU, pediatric ICU; NICU, neonatal ICU; NICU-OH, NICU open-heart surgery; CICU, cardiac ICU; EICU, emergency ICU; GI, gastroenterology; BMT, bone marrow transplant.

**Table 2 jof-08-00183-t002:** Summary AMOVA table. Probability, P(rand ≥ data), for *R_ST_* is based on standard permutation across the full data set. Pops, populations; dF, degrees of freedom; SS, sum of square; MS, mean squares; Est. Var., estimated variance. *p*-value shows no genetic differentiation between the Danish and Iranian isolates. Furthermore, 99% of the variation lies within the two populations.

Source	df	SS	MS	Est. Var.	%
Among Pops	1	268.378	268.378	1.604	1%
Within Pops	160	22,871.048	142.944	142.944	99%
Total	161	23,139.426		144.548	100%
**Stat**	**Value**	**P(rand ≥ data)**			
*R_ST_*	0.011	0.143			

## Data Availability

Deposition of isolates and sequence data. Six C. orthopsilosis isolates obtained from this study were deposited in the Westerdijk Fungal Biodiversity Institute culture collection, Utrecht, the Netherlands, with the accession numbers CBS 15856–15861. Sequences obtained for *ERG11* (MK521929–MK521979) and HS1 and HS2 of *FKS1* (MK522274–MK522386) were deposited in NCBI GenBank.

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
