# Peer review of "Candidemia among Hospitalized Pediatric Patients Caused by Several Clonal Lineages of Candida parapsilosis"

_jof, 2022, doi:10.3390/jof8020183_

Round 1
Reviewer 1 Report
According to me previous review, I think that the manuscript has been improved during the revisions and it might be suitable for publication. Anyway, some limitations remain. I agree with the format of the current version, concerning the use of supplementary material for including all particular details.
Author Response
We appreciate the reviewer agrees with the current version.
Reviewer 2 Report
Dear authors,
though you decided not to present data in comparison with "gold standard", the improvements you made make the article worth publishing. Hope you will get the NGS results too (WGS of closely related strains).
Author Response
We appreciate the reviewer agrees with the current version. WGS of closely related strains may be a potential follow-up study in the future.
Reviewer 3 Report
Dear Authors,
The revised version is significantly improved. However, I believe two conclusions needs to be softened/modified.
Specifically, you stated the AFLP was not appropriate/inferior to STR genotyping for strain typing/cluster identification. I think you need to quantify the statement and specify that the AFLP you are referring to in this study is only the "HpyCH4 IV and MseI primers" you used to generate the fragment patterns, and using the specific electrophoresis system. If you use additional primers targeting more restriction recognition sites and/or use an electrophoretic system capable of providing higher resolution gel pictures (e.g. can differentiate fragments by 1-2bp), you could find more genetic differences among the strains. Similarly, if you had used <5 STR loci, the microsatellite genotyping system might not be able to generate the stated population structure. In contrast, more STR markers could provide greater resolution.
Also, I'm not sure that DAPC can distinguish clonal lineages from recombined genotypes. DAPC maps depict genetic relationships among isolates, showing genotypic clusters but not necessary clonal clusters in the population. Closely related isolates could be siblings from sexual crosses.
Author Response
Dear Authors,
The revised version is significantly improved. However, I believe two conclusions needs to be softened/modified.
1) Specifically, you stated the AFLP was not appropriate/inferior to STR genotyping for strain typing/cluster identification. I think you need to quantify the statement and specify that the AFLP you are referring to in this study is only the "HpyCH4 IV and MseI primers" you used to generate the fragment patterns, and using the specific electrophoresis system. If you use additional primers targeting more restriction recognition sites and/or use an electrophoretic system capable of providing higher resolution gel pictures (e.g. can differentiate fragments by 1-2bp), you could find more genetic differences among the strains. Similarly, if you had used <5 STR loci, the microsatellite genotyping system might not be able to generate the stated population structure. In contrast, more STR markers could provide greater resolution.
Response: revised as suggested. The paragraph now reads (new text is underlined) line 256-265: “This can be explained by the low resolution of the current format of the AFLP that introduce null-alleles resulting in binary data. Moreover, the mutation model behind AFLP markers is based on frequency of mutations in restriction sites, whereas microsatellites has a much higher mutation rate, and a stepwise mutation model that allows inclusion of repeat length. This conclusion was supported by a previous study [14], that found an insufficient resolution by AFLP and suggested that STR should be the preferred method for genetic analysis to uncover outbreaks. Nevertheless, it remains to be understood if inclusion of additional primers targeting more restriction recognition sites and/or use an electrophoretic system capable of providing higher resolution gel pictures would improve the performance of AFLP.”
2) Also, I'm not sure that DAPC can distinguish clonal lineages from recombined genotypes. DAPC maps depict genetic relationships among isolates, showing genotypic clusters but not necessary clonal clusters in the population. Closely related isolates could be siblings from sexual crosses.
Response: revised as suggested. We agree with the reviewer (3) that DAPC cannot distinguish between clonal and recombining populations. What we meant was that the pattern obtained by the DAPC analysis is not in conflict with the IA and d showing clonality, but is what is expected if the fungus has a clonal population structure. This has now been corrected:
Line 254-256 now reads: “The results of the microsatellite analyses showed that the Iranian outbreak that was considered as two clones when assayed with AFLP markers, consisted of five clusters.”
Line 268-269 now reads: “One cluster was only found in Iran, indicating that new clusters may still occur locally.”
Line 270-273 now reads: “The clonal nature of the pathogen is obvious from the analyses of IA and d. Though the DACP analysis cannot distinguish clonal population structures from recombining, the minor overlap between clusters, and little uncertainty in cluster assignment indicate clonality.”
This manuscript is a resubmission of an earlier submission. The following is a list of the peer review reports and author responses from that submission.
Round 1
Reviewer 1 Report
This study investigated whether C. parasilosis isolates from pediatric patients in an Iranian hospital belonged to clonal clusters. The authors used three genotyping methods, AFLP, six microsatellite (simple sequence repeat) markers, and gene sequences at two drug target genes. While the clinical data were clearly presented for these isolates, the presentations of other data, including genotype data analyses were insufficient to support the conclusions. Below are my specific comments/suggestions.
- In the Introduction, the authors should clearly define how clonal clusters will be determined? Generally, a clonal cluster refers to an ancestral genotype and all its clonal descendents, including mutations accumulated during their clonal reproduction. The alternative would be sexual/parasexual recombination that break up the clonal clusters.
- In the Methods section, authors should introduce how they can distinguish those two scenarios using tests of clonality and recombination. The accumulated mutations during clonal reproduction could include those impacting AFLP patterns and microsatellite repeat numbers, and base substitutions within ERG11 and FKS1 genes as revealed by the authors. Appropriate tests should be introduced to determine whether the observed genetic variations is consistent with sexual/parasexual recombination or clonal reproduction to cause the outbreaks.
- Also in the Materials and Methods section, more details should be provided to address objectives stated in the introduction. For example, how do you statistically compare the effectiveness of AFLP and Microsatellite genotyping in discriminating strains and genotypes? One primer pair used for AFLP will most likely be more informative than one microsatellite marker. However, the reverse will likely be true if hundreds of microsatellite markers were used (AFLP does assay many sites within the genome). At the whole genome-sequence level, all isolates will likely be different from each other. I think the authors should state clearly the assumptions and denominators used in their comparison.
- With regard to (in)congruence in data between AFLP and microsatellite markers, it wasn't clear to me what measure(s)/test was used to determine whether the observed incongruence was statistically meaningful. I suggest a Mantel test between two distance matrices between all pairs of strains, one based on AFLP and the other based on microsatellite markers. For microsatellite data, the authors should use Bruvo's distance that takes into account of repeat number differences between strains at each locus.
- The similarity/difference between the Danish and Iranian samples should be statistically tested using established population genetic measures such as Rst, using the microsatellite data.
- I'm unable to see the sequence data for either ERG11 or FKS1 using the provided accession numbers. Those sequences should be released. In addition, since Candida parapsilosis is a diploid, are the mutations mentioned by the authors for these two genes all in homozygous states for all strains?
- The microsatellite genotype data for all strains presented here (including both the Iranian and the Danish data) should be provided in a supplementary table for Data Transparency purpose.
- Minor correction: ERG11 and FKS1 should not be called drug resistance genes. They are drug target genes with certain mutations in them leading to drug resistance.
Author Response
This study investigated whether C. parasilosis isolates from pediatric patients in an Iranian hospital belonged to clonal clusters. The authors used three genotyping methods, AFLP, six microsatellite (simple sequence repeat) markers, and gene sequences at two drug target genes. While the clinical data were clearly presented for these isolates, the presentations of other data, including genotype data analyses were insufficient to support the conclusions. Below are my specific comments/suggestions.
- In the Introduction, the authors should clearly define how clonal clusters will be determined? Generally, a clonal cluster refers to an ancestral genotype and all its clonal descendants, including mutations accumulated during their clonal reproduction. The alternative would be sexual/parasexual recombination that break up the clonal clusters.
Response 1: Revised as requested:
The following sentences have been added to the introduction:
Line 67-69: “As C. parapsilosis is considered asexual this species does not appear to undergo mating and therefore recombination (Butler, Nature 2009). Clusters were thus determined as clonal genotypes and would be expected to dominate in a clonal outbreak.
Additionally: If allowing 1 microsatellite marker difference to be accepted within clonal clusters the results were the same: 19% clusters among Iranian genotypes vs. 17% clusters among Danish genotypes, and 49% of Iranian isolates from different patients in clusters vs. 42% of Danish isolates in clusters. Because clonal outbreaks were not considered here, only identical genotypes were regarded in this study for clarity. This is commented further in Results
Line 157-163: “Since spontaneous mutations occur during clonal reproductions, minor variations in microsatellite repeat numbers (microevolution) could be expected over long time periods yet still represent the same original clone. When performing the same comparisons allowing clusters to comprise also genotypes with 11 of 12 identical markers, the results were the same; 19% clusters among Iranian genotypes vs. 17% clusters among Danish genotypes, and 49% of Iranian isolates from different patients in clusters vs. 42% among Danish isolates.“
- In the Methods section, authors should introduce how they can distinguish those two scenarios using tests of clonality and recombination. The accumulated mutations during clonal reproduction could include those impacting AFLP patterns and microsatellite repeat numbers, and base substitutions within ERG11 and FKS1 genes as revealed by the authors. Appropriate tests should be introduced to determine whether the observed genetic variations is consistent with sexual/parasexual recombination or clonal reproduction to cause the outbreaks.
Response 2: No change as noted above in response to Q1, recombination is absent in C. parapsilosis.
- Also in the Materials and Methods section, more details should be provided to address objectives stated in the introduction. For example, how do you statistically compare the effectiveness of AFLP and Microsatellite genotyping in discriminating strains and genotypes? One primer pair used for AFLP will most likely be more informative than one microsatellite marker. However, the reverse will likely be true if hundreds of microsatellite markers were used (AFLP does assay many sites within the genome). At the whole genome-sequence level, all isolates will likely be different from each other. I think the authors should state clearly the assumptions and denominators used in their comparison.
Response 3: no change: While we agree that statistical comparison is important in cases where differences between methods are minor and statistical analyses may help determining if observed differences can be explained by random variation, we do not see a need in this case. This because the AFLP typing failed to differentiate between the completely unrelated control strains from the Dutch collection (three CBS strains originating from fruit in Italy, plant in Chile and a clinical sample from Puerto Rico in GP-1 and four CBS strains from clinical isolates from Denmark, Norway, Austria and Dom Republic, respectively, in GP-2), and because AFLP also could not differentiate the CBS control strains from the unrelated Tehran isolates. Taking a) this into account as well as b) the fact that three other reviewers did not ask for this calculations, c) the fact that reviewer 4 advised to shorten the ms, and d) the fact that the objective of the study was to confirm or rule out an outbreak and illustrate that the use of AFLP may lead to false conclusions and not to perform a strict method comparison and evaluation, we have not included statistical calculation of the discriminatory power of the two methods used. However, we agree that future studies with systematic comparisons of available typing methods including WGS are warranted.
- With regard to (in) congruence in data between AFLP and microsatellite markers, it wasn't clear to me what measure(s)/test was used to determine whether the observed incongruence was statistically meaningful. I suggest a Mantel test between two distance matrices between all pairs of strains, one based on AFLP and the other based on microsatellite markers. For microsatellite data, the authors should use Bruvo's distance that takes into account of repeat number differences between strains at each locus.
Response 4: no change. Referring to the response to Q3 including we have not included new statistical calculations.
- The similarity/difference between the Danish and Iranian samples should be statistically tested using established population genetic measures such as Rst, using the microsatellite data.
Response 5: no change. Referring to the response to Q3 including we have not included new statistical calculations
- I'm unable to see the sequence data for either ERG11 or FKS1 using the provided accession numbers. Those sequences should be released. In addition, since Candida parapsilosis is a diploid, are the mutations mentioned by the authors for these two genes all in homozygous states for all strains?
Response 6: revised as suggested. Sequences are available on Genbank. Moreover information on homozygosity is clarified, line 166-168, as all mutations were homozygous. The sentence now reads:
“Thirty-eight isolates were ERG11 wild-type and twelve isolates (24%) had one or two homozygous mutations leading to the following amino acid changes: D133Y, Q250K, I302T and R398I.”
- The microsatellite genotype data for all strains presented here (including both the Iranian and the Danish data) should be provided in a supplementary table for Data Transparency purpose.
Response 7: Revised as suggested. Supplementary Table S2 added and referred to in Results line 146:
”Microsatellite typing revealed 37 unique genotypes among the 50 Iranian C. parapsilosis isolates (Figure 1 and Table S2).”
- Minor correction: ERG11 and FKS1 should not be called drug resistance genes. They are drug target genes with certain mutations in them leading to drug resistance.
Response 8: Revised as suggested, line 61-63 now reads:
“In addition, sequencing of drug target genes (primarily ERG11), have previously contributed to delineate clonal outbreaks [6,11,18]. We thus sequenced drug target genes (FKS1 and ERG11)…”

Reviewer 2 Report
The manuscript is clearly presented, conclusions are confirmed by the results and controls are very thorough. The limitations are well stated.
The data support that AFLP is not a preferred method to identify nosocomial spread. The macrosatellite shows sufficient distance to indicate that it is unlikely the Iranian cases were not due to an outbreak, although some patient transfer could not be excluded without knowing the data for patients with identical genotypes shown in Figure 2.
A few minor issues
The figure in Table 1 includes abbreviations not used in the table and does not include some abbreviations used (CHD) - this should be reviewed.
"implies the significant contribution, which microsatellite typing may add in infection control strategies" - assumedly this means that if significant clusters were identifed then infection control studies would have to be put in place - hopefully not too late.
Reviewer 3 Report
Dear authors,
I have evaluated your manuscript “Genotypic studies of a pseudo-outbreak with Candida parapsilosis among hospitalized pediatric patients”. The study is well-written and presents interest to the researchers. What is lacking is the comparison of data with the gold standard in the investigation of outbreaks in wards and hospitals - NGS. It would add to the quality of the manuscript. At least the data on comparison of microsatellite and NGS analysis on strains that had similar patterns in the former both in Denmark and Iran.
Sincerely
Reviewer 4 Report
The main objective of this manuscript is testing which of the two methods of typing methods (Amplified Fragment Length Polymorphism (AFLP) fingerprinting and microsatellite genotyping) is more appropriate for determining if the high percentage of C. parapsilosis were clonal and therefore likely part of a hospital-associated outbreak. A recent study by member of the same group of the current authors (ref. 17) have already suggested that the second method provided a better resolution compared to AFLP fingerprinting, so this study is not really novel, but a confirmation of previous proposal.
In addition, the clinical data (Table 1) and antifungal susceptibility data for the fungal isolates from patients have been published previously, so that the publication of this long and room-consuming Table is reiterative. It would be translated to supp. Material, or just referenced to the previous publication avoid reiteration of publication as a form of self-plagiarism. Fig. 1 should be modified in order to read something. In the current form, it is useless. In turn, it has been already published (ref. 21) by several authors of this new manuscript.
In summary, the manuscript does not contain new data and does not offer new relevant contribution. This new analysis might be published as much as a short letter, but in my opinion is not enough for publishing a new original article. In turn, the final conclusion, at the last three lines and poor and somehow provisional. Discussion emphasizes that there are too limitations yet, due to the low number of samples or certain bias inherent to the comparison between Iran and Denmark.
Author Response
Point 1. The main objective of this manuscript is testing which of the two methods of typing methods (Amplified Fragment Length Polymorphism (AFLP) fingerprinting and microsatellite genotyping) is more appropriate for determining if the high percentage of C. parapsilosis were clonal and therefore likely part of a hospital-associated outbreak. A recent study by member of the same group of the current authors (ref. 17) have already suggested that the second method provided a better resolution compared to AFLP fingerprinting, so this study is not really novel, but a confirmation of previous proposal.
Response 1: No change. Indeed, this study confirms previous proposals arguing that microsatellite typing is more suited for outbreak investigations than AFLP fingerprinting. In contrast to ref. 17, this study applied both AFLP and microsatellite typing and specifically sought to uncover if the isolates were clonal in a separate clinical situation. As AFLP was first applied, an inaccurate suggestion of clonal spread was hypothesized (as in other studies) but later rejected by microsatellite typing. This lead to two important conclusions in this study; first, the Iranian clinical isolates were not part of a clonal outbreak as hypothesized, and second, it is critical to underline the fact that AFLP is unfit for outbreak investigation.
Point 2. In addition, the clinical data (Table 1) and antifungal susceptibility data for the fungal isolates from patients have been published previously, so that the publication of this long and room-consuming Table is reiterative. It would be translated to supp. Material, or just referenced to the previous publication avoid reiteration of publication as a form of self-plagiarism. Fig. 1 should be modified in order to read something. In the current form, it is useless. In turn, it has been already published (ref. 21) by several authors of this new manuscript.
Response 2: Revised accordingly. Table 1 was transferred to Supplementary Table S1. This data has not been presented specifically for each individual patient previously. Fig 1. has not previously been published. The AFLP results in ref. 21 (now 22) corresponds to other C. metapsilosis and C. orthopsilosis isolates from a different study. However, this too, has been moved to Supplementary Figure S1 and replaced with a minor Table 1 (new) summarizing main conclusions from the AFLP analysis along with clinical wards, timespan between isolates grouped by AFLP and which CBS strains that were also found “isogenic” by AFLP, to illustrate and summarize the differences in origin and time of isolates found to share AFLP type.
Point 3. In summary, the manuscript does not contain new data and does not offer new relevant contribution. This new analysis might be published as much as a short letter, but in my opinion is not enough for publishing a new original article. In turn, the final conclusion, at the last three lines are poor and somehow provisional. Discussion emphasizes that there are too limitations yet, due to the low number of samples or certain bias inherent to the comparison between Iran and Denmark.
Response: We succeeded in clarifying and rejecting a hypothesized clonal Candida parapsilosis outbreak at the Teheran pediatric ward in Iran. This has not been presented previously. Furthermore, this study has emphasized the inappropriateness of AFLP for outbreak investigation. This is important as studies exist, misleadingly indicating clonal outbreaks based on AFLP alone and thus relevant for future outbreak studies. The last three lines have been revised, line 231-234:
“In conclusion, our data did not support a hypothesized clonal outbreak at the Teheran pediatric department and stressed that AFLP alone could potentially lead to inaccurate conclusions of clonal outbreaks and should always be supported by other more discriminatory methods.”

Round 2
Reviewer 1 Report
The authors made only minimal and superficial changes to the original version. None of the suggested statistical tests were conducted. Long distance dispersal is common in human fungal pathogens and should not be used as an excuse to not test the incongruence between AFLP and microsatellite markers. Candida parasilosis has NOT been proven to be completely asexual in nature (please read more recent literature published after 2009). In fact incongruence between markers is a signature of sexual/parasexual recombination. If you are not wiling to properly conduct statistical tests as suggested, you can't make any valid conclusions about clonality and differences among markers. Those suggested tests are straightforward to conduct and do not require significant expansion of text. I don't believe Journal of Fungi will refuse to publish a paper because of extra length introduced to provide statistically supported conclusions.
Reviewer 4 Report
The reply letter is mostly convincing and it clarified some of my points of the previous review. The manuscript has been modified according to the criticisms, and the novel points have now been emphasized. The accentuation of the key points discussed at the reply letter has been translated to the amended manuscript. In summary, in my opinion, the manuscript has been improved and the new contributions are clear-cut.